# Pharmacokinetics of Sodium and Calcium Salts of (6S)-5-Methyltetrahydrofolic Acid Compared to Folic Acid and Indirect Comparison of the Two Salts

**DOI:** 10.3390/nu12123623

**Published:** 2020-11-25

**Authors:** Rima Obeid, Christiane Schön, Klaus Pietrzik, Daniel Menzel, Manfred Wilhelm, Yvo Smulders, Jean-Pierre Knapp, Ruth Böhni

**Affiliations:** 1Laboratory Medicine, Department of Clinical Chemistry, Saarland University Hospital, Building 57, D-66424 Homburg/Saar, Germany; 2BioTeSys GmbH, Schelztorstr. 54-56, D-73728 Esslingen, Germany; c.schoen@biotesys.de (C.S.); d.menzel@biotesys.de (D.M.); 3Department of Nutrition and Food Science, Rheinische Friedrich-Wilhelms University, Endenicher Allee 19B, D-53115 Bonn, Germany; k.pietrzik@uni-bonn.de; 4Department of Mathematics, Natural and Economic Sciences, Ulm University of Applied Sciences, Albert-Einstein-Allee 55, D-89081 Ulm, Germany; manfred.wilhelm@thu.de; 5Department of Internal Medicine, Amsterdam UMC, Location VUmc, 1081 HV Amsterdam, The Netherlands; y.smulders@amsterdamumc.nl; 6Merck & Cie, Im Laternenacker 5, CH-8200 Schaffhausen, Switzerland; jean-pierre.knapp@merckgroup.com (J.-P.K.); ruth.boehni@merckgroup.com (R.B.)

**Keywords:** Arcofolin^®^, bioavailability, homocysteine, folic acid, Metafolin^®^, (6S)-5-Methyl-THF, plasma folate, red blood cell folate

## Abstract

(6S)-5-Methyltetrahydrofolic acid ((6S)-5-Methyl-THF) salts and folic acid may differ in their abilities to raise plasma (6S)-5-Methyl-THF levels. We compared the area under the curve (AUC), C_max_, and T_max_ of plasma (6S)-5-Methyl-THF after intakes of (6S)-5-Methyl-THF-Na salt (Arcofolin^®^) and folic acid. Moreover, we compared the AUCs after intakes of (6S)-5-Methyl-THF-Na and the calcium salt, (6S)-5-Methyl-THF-Ca, that were tested against folic acid in two independent studies. The study was randomized, double blind, and cross over. Twenty-four adults (12 men and 12 women) received a single oral dose of 436 µg (6S)-5-Methyl-THF-Na and an equimolar dose of folic acid (400 µg) on two kinetic days with two weeks washout period in between. The plasma concentrations of (6S)-5-Methyl-THF were measured at 9 time points between 0 and 8 h. We found that the AUC_0–8 h_ of plasma (6S)-5-Methyl-THF (mean (SD) = 126.0 (33.6) vs. 56.0 (25.3) nmol/L*h) and C_max_ (36.8 (10.8) vs. 11.1 (4.1) nmol/L) were higher after administration of (6S)-5-Methyl-THF-Na than after the administration of folic acid (*p* < 0.001 for both). These differences were present in men and women. Only administration of folic acid resulted in a transient increase in plasma unmetabolized folic acid (2.5 (2.0) nmol/L after 0.5 h and 4.7 (2.9) nmol/L after 1 h). Intake of (6S)-5-Methyl-THF-Na was safe. The ratios of the AUC_0–8 h_ for (6S)-5-Methyl-THF-Na and (6S)-5-Methyl-THF-Ca to the corresponding folic acid reference group and the delta of these AUC_0–8 h_ did not differ between the studies. In conclusion, a single oral dose of (6S)-5-Methyl-THF-Na caused higher AUC_0–8 h_ and C_max_ of plasma (6S)-5-Methyl-THF compared to folic acid. The Na- and Ca- salts of (6S)-5-Methyl-THF are not likely to differ in their pharmacokinetics. Further studies may investigate whether supplementation of the compounds for a longer time will lead to differences in circulating or intracellular/tissue folate concentrations.

## 1. Introduction

Folates, an essential class of B vitamins, donate one-carbon units during the de-novo synthesis of purines, formylmethionyl-tRNA, thymidylate, serine, and methionine. Thus, folates are needed for synthesis of nucleotides that constitute DNA and RNA structures. In addition, the neurotoxic amino acid homocysteine is converted to methionine using the methyl group of (6S)-5-Methyltetrahydrofolic acid ((6S)-5-Methyl-THF). The methyl group of folate is used to synthesize S-adenosylmethionine that is needed for many methyltransferases (i.e., DNA methyltransferase), thus explaining the wide effects of folate deficiency. Natural sources of folate in the diet are green leafy foods (vegetables), meat, and liver. Dietary (6S)-5-Methyl-THF is conjugated to polyglutamates that are enzymatically removed in the intestine prior to absorption [1]. Folate deficiency is associated with hyperhomocysteinemia, and several diseases such as anemia, depression, and pregnancy complications [2]. Supplementation of folate to women before and during early pregnancy lowers the risk of birth defects [3,4]. Moreover, folates (0.4–0.8 mg/day) lower plasma total homocysteine (tHcy) and may reduce the risk of stroke [5].

(6S)-5-Methyl-THF is the main form of dietary folate and the predominant physiologic folate form in blood [6] and in umbilical cord blood [7,8]. Folic acid is the synthetic and fully oxidized folate form that cannot fulfil physiological functions and is not present in blood unless added to foods or supplements. When supplemented, folic acid is reduced to dihydrofolate then to tetrahydrofolate (possibly in the intestinal wall and the liver) before it is converted to the biologically active form, (6S)-5-Methyl-THF. Detectable levels of unmetabolized folic acid occur temporally in plasma after consumption of > 200 µg folic acid [9]. The meaning of unmetabolized folic acid in plasma is controversially discussed, especially that its concentrations increase parallel to that of total folate in response to folic acid supplementation [10,11]. (6S)-5-Methyl-THF-Ca (i.e., Metafolin^®^) is the calcium salt of (6S)-5-Methyl-THF, a bioactive form of folic acid which has been suggested to be advantageous for instance in individuals with methylenetetrahydrofolate-reductase (MTHFR) 677C>T polymorphism [12]. (6S)-5-Methyl-THF-Ca has been found to be at least as effective as folic acid in raising concentrations of plasma- or red blood cell (RBC)-folate or lowering tHcy levels [13,14,15,16,17]. The Area Under the Curve (AUC) and the C_max_ of plasma total folate have been shown to be higher, and the T_max_ to be shorter after (6S)-5-Methyl-THF-Ca (451 µg) compared to those after an equimolar single dose of folic acid (400 µg) [13]. A recent 12 weeks supplementation trial in women suggested that (6S)-5-Methyl-THF-Ca is superior to folic acid (both 1 mg/day) in raising plasma- and RBC-folate levels [18]. It is advantageous to improve folate status within a short time period (i.e., before closure of the neural tube at 4th week of gestation), such as when women start taking supplements only shortly before or after conception.

Several randomized controlled trials [15,17,19] and pharmacokinetic studies [13] have compared (6S)-5-Methyl-THF-Ca and folic acid in their ability to enhance folate status in women of childbearing age. However, the use of folate supplements is not restricted to women. In a study in male subjects, a single oral dose of (6S)-5-Methyl-THF-Ca and folic acid (both 500 µg) did not differ in the AUC of plasma total folate [16]. Sex-related differences in blood folate markers have been reported [6] which may suggest that men and women could differ in absorption, metabolism, or elimination of folate derivatives.

(6S)-5-Methyl-THF salts may differ in their chemical characteristics (i.e., stability, dissociation rate, and solubility). Therefore, folate salts may also differ in their kinetics and the ability to raise plasma concentrations of (6S)-5-Methyl-THF. (6S)-5-Methyl-THF-Na (Arcofolin^®^) is the monosodium salt of (6S)-5-Methyl-THF (Appendix A). The present study investigated the pharmacokinetics of (6S)-5-Methyl-THF-Na compared to folic acid in adults. We compared the AUC, C_max_, and T_max_ of plasma (6S)-5-Methyl-THF after an equimolar single dose of (6S)-5-Methyl-THF-Na versus folic acid. In addition, we investigated potential differences in the pharmacokinetics of (6S)-5-Methyl-THF-Na and (6S)-5-Methyl-THF-Ca (both tested against folic acid) by conducting pooled data analyses from two independent studies.

## 2. Materials and Methods

### 2.1. (6S)-5-Methyl-THF-Na Pharmacokinetic Study

#### 2.1.1. Subjects and Settings

The present pharmacokinetic study on (6S)-5-Methyl-THF-Na versus folic acid was conducted between October 2018 and February 2019 at an independent Nutritional CRO, BioTeSys GmbH, in Esslingen (Germany). The inclusion criteria were healthy men and women, age between 18 and 50 years, BMI ≥ 18 and ≤ 30 kg/m^2^, plasma folate concentrations between 7 and 45 nmol/L, and RBC-folate concentrations between 405 and 952 nmol/L. The exclusion criteria were smoking, the presence of anemia (hemoglobin <12.0 g/dL in women and 13.0 g/dL in men), vitamin B12 levels <148 pmol/L, plasma tHcy levels ≥15.0 µmol/L, plasma creatinine >0.96 mg/dL for women and >1.21 mg/dL for men, pregnancy, breast-feeding or intention to become pregnant during the study, use of supplements containing folic acid in the last 3 months, regular use of drugs or compounds that may interfere with folate status or assay (i.e., antacids, anticoagulants, omeprazole, methotrexate, vitamin C, high dose biotin, antidepressants, and antiepileptic drugs), diseases potentially interfering with folate absorption or metabolism (i.e., gastro-intestinal diseases, diabetes, cancer, and alcoholism), current infections, cardiovascular events, and marked changes in lifestyle or medications in the last 3 months. The study protocol was reviewed and approved by the Institutional Review Board of Landesärztekammer Baden-Württemberg (approval number: F-2018-077). The study was conducted in accordance to the ethical principles documented in the declaration of Helsinki, and all patients provided their written informed consent to the study. The study is registered at the German Register for Clinical Studies (DRKS-ID: DRKS00015783).

The study test products consisted of 400 µg folic acid (pteroylmonoglutamic acid) and an equimolar amount of (6S)-5-Methyl-THF-Na (436 µg). Both folate compounds were provided in odorless gelatin capsules of identical appearance (Apotheke Roter Ochsen AG, Schaffhausen, Switzerland). The label carried a random ID number and the kinetic day 1 and day 2.

#### 2.1.2. Study Design

This is a randomized, double blind, cross over trial (Appendix A). The participants were recruited via advertisements in local newspapers and public notice boards. Eligibility was first checked through a telephone interview. The study information was sent to the potentially suitable participants. Subjects who were interested in participation were invited for the screening visit at the study center. During the screening visit, detailed information was distributed on the study aims and procedure. Moreover, the fulfilment of the inclusion and exclusion criteria was verified by physical examination, medical history, and blood sampling.

After the screening visit, eligible participants were invited to the study center at two kinetic days. The participants were instructed to maintain their usual diet, avoid alcohol drinks for at least 24 h prior to the study kinetic days, and to consume a light meal in the evening prior to the kinetic days (i.e., farmhouse bread with cream cheese and peeled cucumber). During the study kinetic days 1 and 2, food and beverages other than those provided by study personnel were not permitted.

The randomization was carried out on kinetic day 1 by means of consecutive numbering and was stratified by sex. On the kinetic days, a permanent venous catheter was inserted in the morning. Blood (8 mL) was collected under fasting conditions (at t0) into EDTA-K^+^-containing tubes. The study record diary and compliance documents were handed out. The study product (1 capsule) was administered with 150 mL water by the study personnel to ensure compliance. A folate-free protein drink (Scandishake^®^ Mix, Nutricia GmbH, Erlangen, Germany) was dissolved in water and served at 1 h, 3 h and 6 h after the product intake. The drink provided 430 kcal per serving. On each kinetic day, 9 blood samples were collected between t0 and t8 h (8 mL each). The diary records including documentation of adverse events were returned during the last blood collection session after 8 h of the intake. After a washout period of 2 weeks (only two participants returned at 11 and 17 days) and during the second study visit (kinetic day 2), the other study compound was administered following exactly the same procedure. Figure 1 shows the study flow diagram.

#### 2.1.3. Study Objectives and Sample Size Estimation

The primary objective of the study was to compare the AUC_0–8 h_ of plasma (6S)-5-Methyl-THF between (6S)-5-Methyl-THF-Na and folic acid. The hypothesis was that there is no difference between the AUC_0–8 h_ of (6S)-5-Methyl-THF after intakes of (6S)-5-Methyl-THF-Na versus folic acid.

The secondary objectives were to compare C_max_ and T_max_ of plasma (6S)-5-Methyl-THF between the two folate compounds. C_max_ is the maximum plasma increase of (6S)-5-Methyl-THF achieved after receiving the study compound, and T_max_ is the time needed to reach the C_max_. Furthermore, the same pharmacokinetic parameters (AUC_0–8 h_, C_max_, and T_max_) were also determined for plasma total folate.

The sample size estimation was based on data of a previous independent randomized, double-blind, cross-over pharmacokinetic study conducted at the University of Bonn, Germany. The study compared the incremental AUC_0–8 h_ of plasma total folate after a single equimolar dose of (6S)-5-Methyl-THF-Ca and folic acid in 23 women (Pietrzik et al., 2000 unpublished data). The mean and standard deviation (SD) of AUC_0–8 h_ for folic acid were 50.3 (20.1) nmol/L*h, and the corresponding AUC values for (6S)-5-Methyl-THF-Ca were 85.0 (21.9) nmol/L*h. A priori sample size estimation was based on the difference between the log-transformed means of the AUC_0–8 h_ after (6S)-5-Methyl-THF-Ca versus folic acid (t-test for matched pairs). We estimated that at least 8 participants are needed per group to achieve 90% power (α = 0.05). We planned to recruit 12 women and 12 men to show differences between AUCs of (6S)-5-Methyl-THF-Na and folic acid in a cross over design.

#### 2.1.4. Blood Analyses and Analytical Methods

In order to verify the inclusion and exclusion criteria during the screening visit, blood samples (overnight fasting ≥10 h) were collected into dry tubes, tubes containing EDTA-K^+^, and those containing NaF. The blood samples were centrifuged within 30 min for 10 min at 3000× *g* and 4 °C. Moreover, 1300 µL of 0.5% ascorbic acid aqueous solution were added to 50 µL EDTA-K^+^ whole blood to obtain erythrocyte lysates for RBC-folate assay. EDTA-plasma and serum samples were stored at −70 °C until measurements.

Plasma samples collected during the screening visit were used to measure total folate and vitamin B12 by using Electrochemiluminescence Elecsys^®^ (Cobas, Roche, Mannheim, Germany) and homocysteine by Chemiluminescence Microparticle Assay (Architect system, Abbott Laboratories) (both measured at the Department of Clinical Chemistry, Saarland University Hospital). Folate concentrations were measured in erythrocyte lysates using Chemoluminescence immunoassay (IMMULITE^®^ 2000, Siemens, Munich, Germany) at Synlab Medizinisches Versorgungszentrum (Leinfelden-Echterdingen, Germany). The RBC-folate concentrations were calculated by correcting the concentrations in the erythrocyte lysates by individual hematocrit levels and plasma total folate that were measured in the same time by the IMMULITE^®^ 2000 assay. Blood count and routine markers were performed at Synlab.

During the kinetic days 1 and 2, plasma (6S)-5-Methyl-THF and unmetabolized folic acid were measured using an established ultra-performance liquid chromatography tandem mass-spectrometer (UPLC-MS/MS) method and compound-specific isotope labelled compounds (Merck & Cie, Schaffhausen, Switzerland) at the Department of Clinical Chemistry, Saarland University Hospital, Germany as published before [20]. The between-day coefficient of variation (CV%) for the (6S)-5-Methyl-THF assay was 5.7% at 85 nmol/L and 9.5% at 7.0 nmol/L. The CV% for unmetabolized folic acid was 13.7% at 0.77 nmol/L. The limit of detection and the limit of quantification for folic acid in plasma were 0.20 nmol/L and 0.40 nmol/L, respectively. Plasma total folate levels were measured on the kinetic days by using Cobas autoanalyser (Elecsys^®^, Roche, Mannheim, Germany) at the Department of Clinical Chemistry, Saarland University Hospital, Germany. The CV% of the total folate assay was <6%. All plasma samples collected from the same subject (18 samples on both study visits) were prepared and measured in the same run. All blood samples were primary aliquots frozen at −70 °C for no longer than 3 months.

#### 2.1.5. Statistical Analyses

The pharmacokinetic endpoints (AUC_0–8 h_, C_max_ and T_max_) were calculated from individual concentration-time curves. Incremental areas under the observed concentration-time curve above baseline were calculated applying the trapezoidal rule and were expressed as nmol/L*h.

The linear mixed model was applied using the log-transformed data to investigate the differences in AUC_0–8 h_ of (6S)-5-Methyl-THF (the primary endpoint) between (6S)-5-Methyl-THF-Na and folic acid groups taking into account sequence of the study products, kinetic days (1 or 2), and the intervention with (6S)-5-Methyl-THF-Na or folic acid.

The difference in the AUC_0–8 h_ of plasma (6S)-5-Methyl-THF between the study products was additionally investigated after stratification by sex. Further post-hoc subgroup analyses included analyses of the results according to strata of concentrations of total folate, RBC-folate, vitamin B12, homocysteine, and hemoglobin as measured during the screening visit. The subgroups were stratified by median values of the biomarkers in the whole group (≤ and > median), except for hemoglobin that was stratified by the median value in women and men separately. For the subgroup analyses, the interaction between “intervention and subgroup” was studied. When the interaction was significant, the subgroup was considered as a covariate in the linear mixed model and the differences in AUC_0–8 h_ of (6S)-5-Methyl-THF were additionally investigated within the subgroups. The same test statistic as described for the primary endpoint was applied for C_max_. Differences between T_max_ were evaluated by Wilcoxon rank sum test.

Statistical analyses were conducted using the GraphPad Prism (Version 5.06), SPSS (IBM, Version 24.0), SAS (9.3), and Microsoft Excel statistical packages. Continuous variables are shown as mean (SD) or median (25th-75th percentiles).

### 2.2. AUCs for (6S)-5-Methyl-THF-Ca Versus Folic Acid and Methods of Indirect Comparisons with the AUCs for (6S)-5-Methyl-THF-Na

Prinz-Langenohl et al. (2009) compared the AUC_0–8 h_ of plasma total folate between (6S)-5-Methyl-THF-Ca and folic acid in 24 young women [13]. Appendix A shows a brief summary of the study design compared to the present study. Equimolar oral doses of (6S)-5-Methyl-THF-Na (436 µg, present study) and (6S)-5-Methyl-THF-Ca (451 µg, in Prinz-Langenohl et al.) have been tested against the same reference substance, folic acid (400 µg). Plasma concentrations of (6S)-5-Methyl-THF were selectively measured in both studies. In the study by Prinz-Langenohl et al., 2009 [13], the measurements of plasma (6S)-5-Methyl-THF were conducted at VU University Medical Centre, Department of Clinical Chemistry, Amsterdam (The Netherlands), by using liquid chromatography tandem mass-spectrometer (LC-MS-MS system) using ^13^C_5_-(6S)-5-Methyl-THF as internal standard [21]. We here show novel data on the AUC_0–8 h_ of plasma (6S)-5-Methyl-THF from the original study [13].

#### Methods of Indirect Comparisons between Na and Ca Salts of 5-Methyl-THF

Appendix A shows a schematic summary of the indirect comparisons of AUCs of plasma (6S)-5-Methyl-THF for (6S)-5-Methyl-THF-Na and (6S)-5-Methyl-THF-Ca. We used individual data from the two studies to calculate the AUC_0–8 h_ of plasma (6S)-5-Methyl-THF. The blood sampling time points were not identical in the present study and the 2009 study. Sampling at 0.25 h is missing in 2009, and sampling at 1.5 h is missing in 2019. We extrapolated plasma concentrations of (6S)-5-Methyl-THF for the missing time points in the corresponding studies by calculating the mean of the concentrations at the two neighboring time points (=before and after the missing concentrations). The AUCs from the 2019 and 2009 studies were calculated from 10 time points (0, 0.25 h, 0.5 h, 1 h, 1.5 h, 2 h, 3 h, 4 h, 6 h, and 8 h). The incremental AUCs of plasma (6S)-5-Methyl-THF were calculated and were expressed as nmol/L*h.

The AUCs of plasma (6S)-5-Methyl-THF were compared between the 2019 and 2009 studies using t-tests for independent groups. We planned to compare the AUCs between (6S)-5-Methyl-THF-Na 2019 and (6S)-5-Methyl-THF-Ca 2009 if the AUCs for the corresponding reference groups (i.e., folic acid 2019 and folic acid 2009) are not different (lower or higher). However, the AUCs of plasma (6S)-5-Methyl-THF for folic acid were systematically higher in 2009 than those for folic acid in 2019 (Appendix A). Therefore, we undertook a correction step by subtracting 17 nmol/L*h “the difference in AUCs between the reference substance in 2009 and 2019” from all individual AUCs for (6S)-5-Methyl-THF-Ca 2009 (Appendix A).

Additionally, we compared the ratio of the native AUCs of plasma (6S)-5-Methyl-THF for (6S)-5-Methyl-THF-Na/ to the AUC folic acid 2019 versus the ratio of AUCs for (6S)-5-Methyl-THF-Ca/to the AUC for folic acid 2009 and the differences between the study-specific uncorrected AUCs (i.e., the AUC for folic acid minus the AUC for the folate salt) (Appendix A). We further evaluated the 90% confidence interval (CI) of the ratio of the geometric means of AUC_0–8 h_ for (6S)-5-Methyl-THF-Na to the corrected AUC_0–8 h_ for (6S)-5-Methyl-THF-Ca. (6S)-5-Methyl-THF-Na and (6S)-5-Methyl-THF-Ca were considered bioequivalent, if the 90% CI of the ratio of geometric means of AUC_0–8 h_ were between 0.80 and 1.25 (i.e., the limits defined by the FDA for bioequivalence). Additional tests were conducted to proof the comparability and exchangeability of the folate analytical methods between the (6S)-5-Methyl-THF-Na and (6S)-5-Methyl-THF-Ca pharmacokinetic studies (Appendix A).

The data analyses were conducted using individual data from the 24 subjects in the present pharmacokinetic study and 21 women (3 women were excluded due to implausible results) from the study of Prinz-Langenohl et al. [13].

## 3. Results

### 3.1. (6S)-5-Methyl-THF-Na Pharmacokinetic Study

#### 3.1.1. Baseline Characteristics and Folate Markers during the Screening Visit

The present study included 24 participants (mean (SD) age = 29.7 (7.5) years; 12 men and 12 women). All subjects completed both kinetic days. Mean and SD of plasma tHcy, plasma vitamin B12, plasma folate, and RBC-folate are shown in Table 1.

#### 3.1.2. Changes of Plasma Concentrations of (6S)-5-Methyl-THF and Total Folate

Plasma concentrations of (6S)-5-Methyl-THF increased from t0 to 8 h after both (6S)-5-Methyl-THF-Na and folic acid (Table 2, Figure 2A). The concentrations of unmetabolized folic acid in plasma were below the limit of detection in all samples before the intervention. After the intake of folic acid, the concentrations of unmetabolized folic acid increased from below the limit of detection in all participants to a mean (SD) of 2.51 (2.03) nmol/L at 0.5 h, and to 4.72 (2.90) nmol/L at 1 h. In contrast, unmetabolized folic levels remained below the limit of detection after the intake of (6S)-5-Methyl-THF-Na (Figure 2C). Moreover, the concentrations of total folate in plasma showed a more marked increase after (6S)-5-Methyl-THF-Na than after folic acid (Table 2, Figure 2B).

#### 3.1.3. Pharmacokinetic Markers after (6S)-5-Methyl-THF-Na and Folic Acid Intake

Plasma concentration of (6S)-5-Methyl-THF was used to calculate the AUC_0–8 h_ (the primary outcome). The mean AUC_0–8 h_ after the intake of (6S)-5-Methyl-THF-Na was significantly higher than that after folic acid (mean (SD) = 126.0 (33.6) vs. 56.0 (25.3) nmol/L*h; *p* < 0.0001) (Table 3). Further, the secondary endpoints, C_max_ and T_max_ of plasma (6S)-5-Methyl-THF, were also significantly different between the study products (Table 3). The C_max_ was higher after the intake of (6S)-5-Methyl-THF-Na compared to that after folic acid (36.8 (10.8) nmol/L vs. 11.1 (4.1) nmol/L, respectively; *p* < 0.0001). The T_max_ after the intake of (6S)-5-Methyl-THF-Na (median (25th–75th Percentile) = 60 (30–60) min, range: 30–240 min) was shorter than the T_max_ after folic acid (120 (120–165) min, range: 60–480 min; *p* = 0.0002).

The AUC_0–8 h_ of plasma total folate was significantly higher after the intake of (6S)-5-Methyl-THF-Na than that after folic acid (89.2 (19.1) nmol/L*h vs. 71.1 (17.8) nmol/L*h; *p* = 0.0007). The C_max_ tended to be higher after (6S)-5-Methyl-THF-Na compared to that after folic acid (23.9 (7.3) nmol/L vs. 20.7 (7.7) nmol/L; *p* = 0.0564). The T_max_ tended to be shorter after the intake of (6S)-5-Methyl-THF-Na (median (25th–75th Percentile) = 30 (30–60) min, range: 30–360 min) compared to that after folic acid (60 (60–60) min, range: 30–180 min; *p* = 0.0534). Two subjects showed a delayed T_max_ of total plasma folate after intake of (6S)-5-Methyl-THF-Na (360 min and 180 min compared to the mean of all other participants = 41 min). After excluding these two outliers, the maximum plasma concentrations of total folate were found to be reached faster after intake of (6S)-5-Methyl-THF-Na than after folic acid (*p* = 0.0026) (Table 3).

#### 3.1.4. Subgroup Analyses of AUCs of Plasma (6S)-5-Methyl-THF after Intake of (6S)-5-Methyl-THF-Na and Folic Acid

In all subgroup analyses that were conducted, the AUC_0–8 h_ of plasma (6S)-5-Methyl-THF were higher after (6S)-5-Methyl-THF-Na compared to those after folic acid intake (Table 4). There were no significant interactions between the respective study products and any of the subgroups, except for a trend towards interaction between plasma vitamin B12 levels and the study products (*p* = 0.0503). The AUC_0–8 h_ of plasma (6S)-5-Methyl-THF was significantly higher in subjects with vitamin B12 concentrations ≤254 pmol/L compared to those with vitamin B12 levels above this value (*p* = 0.0132).

Differences in AUC_0–8 h_ of plasma (6S)-5-Methyl-THF between (6S)-5-Methyl-THF-Na and folic acid were confirmed in men and women. In an explorative analysis, we compared the AUC_0–8 h_ of plasma (6S)-5-Methyl-THF between men and women. The AUC_0–8 h_ of plasma (6S)-5-Methyl-THF did not differ between women and men after intake of folic acid. After intake of (6S)-5-Methyl-THF-Na, the AUC_0–8 h_ of plasma (6S)-5-Methyl-THF was higher in women compared to men (t-test for independent groups *p* = 0.0366). The sex-differences in AUC_0–8 h_ of plasma (6S)-5-Methyl-THF after (6S)-5-Methyl-THF-Na were no longer significant after adjustment for hemoglobin and plasma vitamin B12 (both were lower in women) (*p* = 0.3736).

#### 3.1.5. Safety

There were no serious adverse events over the study period. Three subjects reported headaches on kinetic day 1 after intake of (6S)-5-Methyl-THF-Na, which, however, were rated as unrelated to the study product. Five participants reported 5 adverse events during the wash-out phase (3 reported a common cold, 1 subject suffered from headache and one subject reported migraine).

### 3.2. The AUCs of Plasma (6S)-5-Methyl-THF after (6S)-5-Methyl-THF-Ca Versus Folic Acid

Figure 3 shows novel data on plasma (6S)-5-Methyl-THF concentrations and the corresponding AUC_0–8 h_ after (6S)-5-Methyl-THF-Ca and folic acid intake from 21 women that participated in the study of Prinz-Langenohl et al. [13]. The AUC_0–8 h_ of plasma (6S)-5-Methyl-THF was significantly higher after (6S)-5-Methyl-THF-Ca compared to that after folic acid (152.9 (36.5) nmol/L*h vs. 73.1 (26.5) nmol/L*h, respectively; *p* <0.0001).

### 3.3. Indirect Comparison of the AUCs between the (6S)-5-Methyl-THF Salts

The changes of plasma concentrations of (6S)-5-Methyl-THF from baseline were systematically higher in the 2009 study compared to the present study (Appendix A).

We observed higher incremental AUC_0–8 h_ of plasma (6S)-5-Methyl-THF for the folic acid group in the 2009-study compared to the folic acid group in the 2019-study (mean (SD) AUC = 73.1 (26.5) nmol/L*h vs 56.0 (25.3) nmol/L*h, respectively; *p* = 0.0203 (unpaired t-test after log-transformation)) (Appendix A). The difference and (95%CI) of the AUCs of the folic acid groups in 2019 and 2009 were 17.0 (1.43, 32.6) nmol/L*h. In the next step, the individual AUCs in 2009 were corrected for this systematic difference by subtracting 17.0 nmol/L*h from all observed AUC values for (6S)-5-Methyl-THF-Ca. After this correction step, the AUC_0–8 h_ of plasma (6S)-5-Methyl-THF were not different for (6S)-5-Methyl-THF-Na and (6S)-5-Methyl-THF-Ca (126.0 (33.6) vs. 135.9 (36.5) nmol/L*h, respectively; *p* = 0.3675 unpaired t-test applied on log-AUCs) (Appendix A). Moreover, as the 90% CI (= 0.93 (0.80, 1.07) of the ratio of the geometric means of AUC_0–8 h_ were between 0.80 and 1.25, the two salts can be considered bioequivalent.

The individual ratios and differences between the AUCs of plasma (6S)-5-Methyl-THF were calculated from both studies as explained in Appendix A. These study-specific measures are internally valid and robust against possible between-study variations in analytical methods, subject characteristics, and experimental conditions. We compared the ratio of the AUCs of plasma (6S)-5-Methyl-THF for (6S)-5-Methyl-THF-Na/AUC folic acid 2019 versus the ratio of AUCs for (6S)-5-Methyl-THF-Ca/folic acid 2009, and the differences between the study-specific AUCs (i.e., the AUC for folic acid minus the AUC for the folate salt). The ratios of the AUCs of plasma (6S)-5-Methyl-THF (*p* = 0.4392) and the differences between the AUCs (*p* = 0.2986) did not differ between the 2019 and 2009 studies (Appendix A).

Additional statistical analyses including comparisons of the analytical methods were performed to ensure the validity of the indirect comparison of AUCs of plasma (6S)-5-Methyl-THF after intake of (6S)-5-Methyl-THF-Na and (6S)-5-Methyl-THF-Ca (Appendix A).

## 4. Discussion

We compared the pharmacokinetic parameters (AUC, C_max_, and T_max_) of plasma (6S)-5-Methyl-THF after (6S)-5-Methyl-THF-Na versus those after an equimolar single dose of folic acid in a group of 24 adults. The AUC_0–8 h_ of plasma (6S)-5-Methyl-THF was significantly higher after (6S)-5-Methyl-THF-Na compared to that after folic acid. In this short-term study, the increase in plasma concentrations of (6S)-5-Methyl-THF was significantly higher after (6S)-5-Methyl-THF-Na than after folic acid. In addition, T_max_ after the intake of (6S)-5-Methyl-THF-Na was significantly shorter than that after folic acid which could be due to a delay in processing folic acid to (6S)-5-Methyl-THF in the liver [22]. The AUC, C_max_, and T_max_ of plasma (6S)-5-Methyl-THF after (6S)-5-Methyl-THF-Ca or folic acid confirmed pervious results of plasma total folate (14), but the differences were stronger for (6S)-5-Methyl-THF. The larger differences in the AUC and C_max_ shown when using the mass-spectrometry analytical method compared to the immunological method could be due to the high selectiveness of the mass-spectrometry in detecting methyl folate (the main folate form in plasma). The significant differences in AUC between (6S)-5-Methyl-THF-Na and folic acid were confirmed among men and women. Plasma concentrations of unmetabolized folic acid reached their peak concentrations after 30–60 min and were undetectable after 8 h of the oral intake of folic acid but remained undetectable after (6S)-5-Methyl-THF-Na.

Supplementation of folate in women of childbearing age aims at achieving protective RBC-folate concentrations (>906 nmol/L) and thereby reducing the risk of some birth defects [23]. However, several factors can delay the raise in blood or plasma folate in women who use supplemental folic acid. For example, homozygosity for the MTHFRC677T polymorphism (TT) is associated with lower plasma- and RBC-folate concentrations in women who use 400 µg/day folic acid [24]. Using (6S)-5-Methyl-THF-salts aims at providing directly the physiological form of folate which has better absorption and can support the folate cycle without the need for enzymatic processing (i.e., via dihydrofolate-reductase (DHFR)). It is further of note that (6S)-5-Methyl-THF, but not folic acid, is actively transported and accumulated in cord blood and the fetus [7,8].

A single dose of (6S)-5-Methyl-THF-Ca also caused higher increase of plasma total folate [13] and plasma (6S)-5-Methyl-THF (present study) compared to folic acid. This may appear to contradict previously reported results on bioequivalence of a single oral dose of (6S)-5-Methyl-THF-Ca and folic acid (each 500 µg) tested in men who were pre-loaded with 5 mg/day folic acid [16]. Plasma total folate (measured by a microbiological assay), but not selectively (6S)-5-Methyl-THF, was used to calculate the AUCs in that study [16]. Moreover, loading with a high dose folic acid may cause saturation of the folate binding protein in the liver, and thus, (6S)-5-Methyl-THF-Ca and folic acid could become less different in their pharmacokinetics in subjects previously exposed to folic acid.

Long term supplementation of (6S)-5-Methyl-THF-Ca in women (i.e., non-pregnant, or lactating mothers) leads to higher RBC-folate and serum folate compared to the concentrations obtained after folic acid [18,25]. A recent randomized-double blind controlled trial in infants has shown that RBC-folate concentrations were higher at 3 months follow up in the group that received milk fortified with (6S)-5-Methyl-THF-Ca compared to the group that received milk fortified with folic acid, although plasma (6S)-5-Methyl-THF did not differ between the groups [26]. It might be argued that folic acid is readily assimilated by tissues as it is reduced to tetrahydrofolate and trapped in the cells after polyglutamation, whereas (6S)-5-Methyl-THF needs to be converted to tetrahydrofolate before being stored. At present, it is not known whether the storage folate marker (i.e., RBC-folate) may differ after long term supplementation of (6S)-5-Methyl-THF-Na versus folic acid. Faster achievement of protective RBC-folate concentrations is advantageous in women planning a pregnancy.

We found that women had a higher mean AUC after (6S)-5-Methyl-THF-Na compared to men. This observation could be due to differences in vitamin B12 and hemoglobin or to physiological differences between men and women in handling folates. Differences in one-carbon metabolism between men and women of childbearing age have been reported before [27] and could be linked to the observed sex-related differences in folate pharmacokinetics. Future studies should confirm these results.

Intake of (6S)-5-Methyl-THF salts may have advantages over intake of folic acid. High dose folic acid may cause inhibition of the MTHFR gene [28]. The rate of conversion of folic acid to tetrahydrofolate by DHFR is 850 times lower than the rate of conversion of 7,8-dihydrofolate, thus potentially limiting metabolism of high doses of folic acid [29]. Indeed, DHFR has been a target of cancer treatment, thus highlighting the essential role of this enzyme in delivering tetrahydrofolates to the cell [30]. Inhibition of DHFR by high dose folic acid may have relevance to birth outcome in women receiving (>1 mg) folic acid during pregnancy. A 19-base pair deletion polymorphism in DHFR (DHFR 19-bp deletion) is common in the population (i.e., 45%) [31] and can influence the efficiency of the enzyme in converting folic acid to tetrahydrofolate, especially when a high dose of folic acid is supplemented [32]. Folic acid at a dose ≥500 µg caused a 2-fold higher prevalence of detectable unmetabolized folic acid in subjects with DHFR 19-bp deletion compared to those with the wild type variant [32]. Even at an intake of folic acid <250 µg/day, the deletion genotype was associated with 100 nmol/L lower RBC-folate levels compared to the wild type genotype [32]. The DHFR 19-bp deletion and folate status shows interaction in the association with memory function in elderly people [33], suggesting that (6S)-5-Methyl-THF salts could be the preferred folate form in elderly people.

We speculate that using (6S)-5-Methyl-THF salts instead of folic acid may be associated with a higher risk reduction of neural tube defects or may in theory prevent cases not prevented by folic acid. This view is supported by data showing that (6S)-5-Methyl-THF-Ca (versus the same dose of folic acid) is associated with higher folate concentrations in blood after 12 weeks [18] and that (6S)-5-Methyl-THF-Ca (vs. folic acid) can maintain RBC-folate after discontinuing folic acid (1 mg/day) [25,34]. The Na- ad Ca- salts of (6S)-5-Methyl-THF enter the folate cycle without the need for DHFR or MTHFR activities. Bypassing DHFR and MTHFR makes (6S)-5-Methyl-THF available for the methionine synthase and other purine and thymidylate biosynthesis pathways. Genetic polymorphisms in genes involved in one carbon metabolism are also associated with vascular disease, cognitive impairment and depression [35,36,37]. Therefore, future studies may investigate whether carriers of these genetic variants may show more benefit from (6S)-5-Methyl-THF salts compared to folic acid.

(6S)-5-Methyl-THF-Na and (6S)-5-Methyl-THF-Ca could theoretically differ in their pharmacokinetics due to their chemical properties, thus differentially affecting the intestinal absorption, cellular storage, or elimination of folate from the circulation. The indirect comparisons of the AUCs of plasma (6S)-5-Methyl-THF between (6S)-5-Methyl-THF-Na and (6S)-5-Methyl-THF-Ca (both tested against folic acid) have shown that the calcium and monosodium salts are not likely to differ in their pharmacokinetics. The indirect comparison of the two salts has some limitations due to using different analytical methods and statistical tests for independent groups to investigate the differences in AUCs that could require a larger sample size to overcome inter-individual and sex-related variations. Systematically higher AUCs after folic acid in 2009 versus 2019 could be due to characteristics of the participants such as including only women and only MTHFR677 TT and CC genotypes in 2009, having different baseline folate concentrations and/or using different analytical methods. However, we used several validation steps including prediction of plasma (6S)-5-Methyl-THF from total plasma folate by using regression models. The ratios and the differences of AUC of (6S)-5-Methyl-THF-Na and (6S)-5-Methyl-THF-Ca to the study-specific AUC for folic acid are valid within a given study and robust to systematic differences.

## 5. Conclusions

The AUC_0–8 h_ and the C_max_ of plasma (6S)-5-Methyl-THF were higher after a single oral dose of (6S)-5-Methyl-THF-Na compared to folic acid. The maximum concentrations of plasma (6S)-5-Methyl-THF were reached 60 min earlier after (6S)-5-Methyl-THF-Na compared to folic acid. In contrast to folic acid, (6S)-5-Methyl-THF-Na did not cause an increase in plasma concentrations of unmetabolized folic acid. Comparing the AUCs of (6S)-5-Methyl-THF-Na and (6S)-5-Methyl-THF-Ca from two independent studies has shown that the two salts are not likely to differ in their pharmacokinetics after a single oral dose. The results of the present study cannot be extrapolated to long-term studies or linked to any clinical outcome. Future studies should investigate the effect of long-term supplementation of (6S)-5-Methyl-THF-Na versus (6S)-5-Methyl-THF-Ca and folic acid on intracellular folate levels (RBC-folate), biochemical markers (i.e., lowering tHcy), and clinical endpoints such as correction of anemia especially in relation to the presence of polymorphisms in the MTHFR and DHFR genes.

## Figures and Tables

**Figure 1 nutrients-12-03623-f001:**
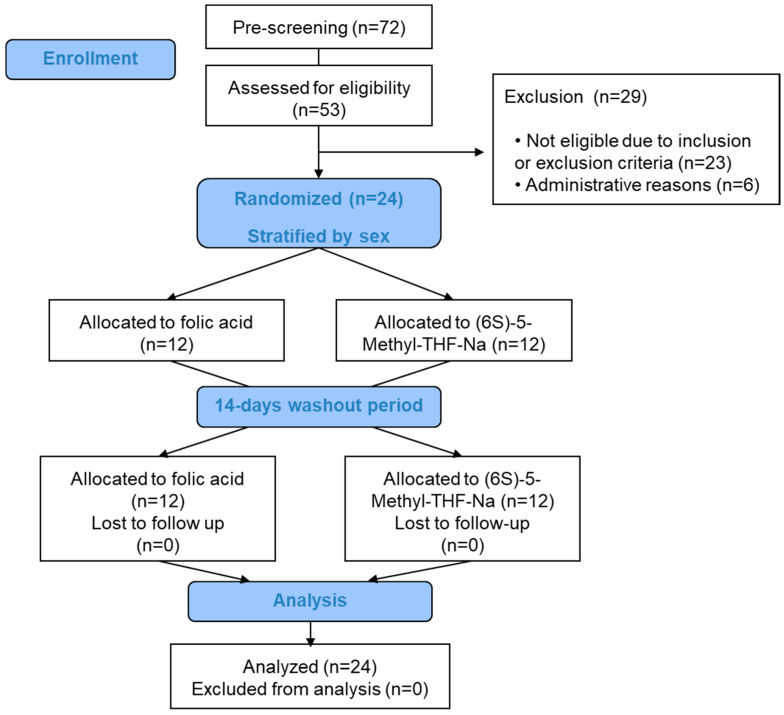
Study flow diagram.

**Figure 2 nutrients-12-03623-f002:**
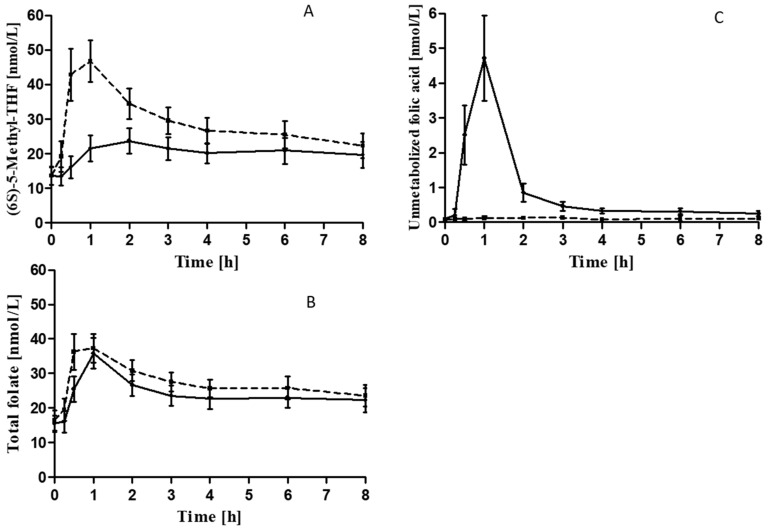
Mean (95%CI) of plasma concentrations of (6S)-5-Methyl-THF (**A**), total folate (**B**), and unmetabolized folic acid (**C**) after intake of (6S)-5-Methyl-THF-Na (dashed line) and folic acid (solid line) over 8 h.

**Figure 3 nutrients-12-03623-f003:**
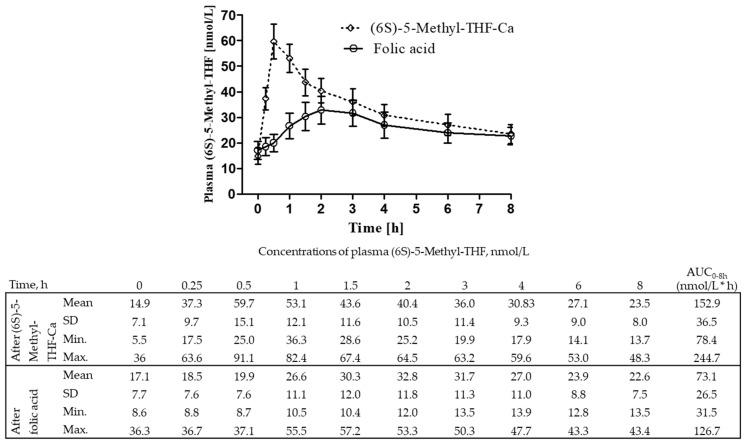
Plasma concentrations of (6S)-5-Methyl-THF after (6S)-5-Methyl-THF-Ca versus folic acid (data in the figure are mean ± 95% CI; n = 21). Concentrations at 0.25 h were estimated as mean concentrations of t0 + t0.5 h. Prinz-Langenohl study [13] (unpublished results).

**Table 1 nutrients-12-03623-t001:** Population characteristics and blood markers during the screening visit.

	All	Men	Women
N	24	12	12
Age, years	29.7 (7.5)	32.1 (9.2)	27.3 (4.6)
BMI, kg/m^2^	23.1 (2.8)	23.6 (2.6)	22.6 (3.0)
Systolic blood pressure, mmHg	121 (16)	130 (10)	112 (15)
Diastolic blood pressure, mmHg	74 (10)	77 (10)	71 (9)
Heart rate, bpm	70 (9)	71 (7)	69 (10)
Hemoglobin, g/dl	14.3 (1.3)	15.2 (0.8)	13.4 (1.0)
tHcy, µmol/L	7.6 (1.8)	7.8 (1.4)	7.4 (2.1)
Vitamin B12, pmol/L	271 (80)	291 (86)	251 (70)
Plasma folate, nmol/L	15.9 (6.0)	14.9 (3.7)	16.9 (7.6)
RBC-folate, nmol/L	664 (174)	669 (183)	658 (173)

Data are mean (SD). BMI, body mass index; tHcy, total homocysteine; RBC-folate, red blood cell folate.

**Table 2 nutrients-12-03623-t002:** Concentrations of plasma (6S)-5-Methyl-THF and total folate and their changes from t0 to 8 h (all in nmol/L) following a single oral dose of (6S)-5-Methyl-THF-Na or folic acid.

Time, h	0	0.25	0.5	1	2	3	4	6	8
**Test compound: (6S)-5-Methyl-THF-Na, 436 µg**									
(6S)-5-Methyl-THF	13.6 (6.4)	19.2 (10.5)	42.8 (17.9)	46.8 (14.2)	34.5 (10.5)	39.6 (9.3)	26.6 (9.0)	25.5 (9.3)	22.3 (8.4)
Change from t0	/	5.6 (6.2)	29.2 (14.1)	33.2 (10.2)	20.8 (5.7)	16.0 (4.5)	13.0 (4.2)	11.9 (4.3)	8.7 (3.8)
Total folate	16.4 (6.7)	19.6 (7.7)	36.3 (12.3)	37.3 (9.7)	30.8 (7.1)	27.6 (6.4)	25.7 (6.1)	25.8 (7.7)	23.6 (7.3)
Change from t0 ^a^	/	3.1 (4.2)	19.9 (9.4)	20.8 (7.3)	14.4 (3.8)	11.1 (3.2)	9.3 (3.5)	9.4 (2.3)	7.8 (2.7)
**Reference compound: folic acid, 400 µg**									
(6S)-5-Methyl-THF	13.5 (6.0)	13.4 (6.3)	16.1 (7.6)	21.5 (9.1)	23.7 (8.8)	21.5 (7.8)	20.1 (7.0)	20.9 (8.9)	19.7 (8.9)
Change from t0	/	−0.1 (2.3)	2.6 (3.2)	8.0 (4.0)	10.1 (3.9)	8.0 (3.0)	6.6 (2.3)	7.3 (4.9)	6.2 (4.5)
Total folate	15.5 (5.6)	16.1 (7.5)	25.5 (8.8)	35.8 (10.5)	26.7 (7.3)	23.6 (7.1)	22.8 (7.1)	23.0 (6.9)	22.3 (8.1)
Change from t0 ^a^	/	0.6 (2.7)	10.0 (6.1)	20.3 (8.1)	11.2 (3.2)	8.1 (2.8)	7.3 (2.6)	7.5 (3.6)	6.8 (3.8)

Results are shown as mean (SD). N = 24 participants (12 men and 12 women). ^a^ The changes of the folate concentrations from t0 are calculated as: concentrations at tx—concentrations at t0.

**Table 3 nutrients-12-03623-t003:** Pharmacokinetic parameters of plasma (6S)-5-Methyl-THF and total folate after oral intake of (6S)-5-Methyl-THF-Na and folic acid (n = 24).

	Folate Compound		
	(6S)-5-Methyl-THF-Na	Folic Acid	*p*
**Plasma (6S)-5-Methyl-THF**			
AUC_0–8 h_, nmol/L*h, mean (SD)	126.0 (33.6)	56.0 (25.3)	<0.0001
C_max_, nmol/L, mean (SD) and (range)	36.8 (10.8) (14.4, 53.0)	11.1 (4.1) (5.3, 24.7)	<0.0001
T_max_, minutes, median (25th–75th Percentiles) (range)	60 (30–60) (30, 240)	120 (120–165) (60, 480)	0.0002 ^a^
**Plasma total folate**			
AUC_0–8 h_, nmol/L*h, mean (SD)	89.2 (19.1)	71.1 (17.8)	0.0007
AUC_0–8 h_, nmol/L*h, mean (SD)	91.6 (18.0)	69.9 (17.3)	<0.0001 ^b^
C_max_, nmol/L, mean (SD) and (range)	23.9 (7.3) (12.1, 42.2)	20.7 (7.7) (7.5, 33.0)	0.0564
C_max_, nmol/L, mean (SD) and (range)	24.9 (6.7) (16.0, 42.2)	20.3 (7.8) (7.5, 33.0)	0.0074 ^b^
T_max_, minutes, median (25th–75th Percentiles) (range)	30 (30–60) (30, 360)	60 (60–60) (30, 180)	0.0534 ^a^
T_max_, minutes, median (25th–75th Percentile) (range)	30 (30–60) (30, 60)	60 (60–60) (30, 180)	0.0026 ^a,b^

Data are shown as mean (SD) or median (25th–75th) percentiles and (range). *p* values are according to the linear mixed model with log-transformed data for AUC and C_max_ taking into account sequence, period, and intervention type (n = 24). ^a^ A non-parametric test was used to compare T_max_. ^b^ Comparisons were repeated after excluding 2 outliers with high T_max_ (extended response) to (6S)-5-Methyl-THF-Na (total n = 22).

**Table 4 nutrients-12-03623-t004:** Subgroup analyses of the AUC of plasma (6S)-5-Methyl-THF after (6S)-5-Methyl-THF-Na and folic acid.

	AUC_0–8 h_ of (6S)-5-Methyl-THF [nmol/L*h]		*p* for AUCs between the Interventions ^a^	*p* for Interaction Product × Subgroup ^b^
	(6S)-5-Methyl-THF-Na	Folic acid		
Men	111.9 (28.9)	51.6 (18.5)	<0.0001	
Women	140.1 (33.0)	60.5 (30.9)	<0.0001	0.4938
**Post-hoc subgroup analyses according to biomarker concentrations at screening ^c^**				
Plasma folate low	125.7 (28.9)	54.7 (21.9)	<0.0001	
Plasma folate high	126.3 (39.1)	57.4 (29.2)	<0.0001	0.8040
RBC-folate low	121.0 (34.4)	54.5 (21.9)	0.0002	
RBC-folate high	130.9 (33.6)	57.5 (29.2)	<0.0001	0.6833
Vitamin B12 low	142.4 (30.0)	57.2 (31.2)	<0.0001	
Vitamin B12 high	109.6 (29.6)	54.9 (19.1)	<0.0001	0.0503
tHcy low	127.6 (40.2)	58.1 (29.5)	<0.0001	
tHcy high	124.3 (27.2)	54.0 (21.4)	<0.0001	0.7090
Hemoglobin low	129.6 (42.2)	57.7 (30.8)	<0.0001	
Hemoglobin high	122.3 (23.6)	54.4 (19.6)	<0.0001	0.9101

Data are shown as mean (SD). ^a^
*p* values are from the linear mixed model including AUC as a dependent variable, the treatment ((6S)-5-Methyl-THF-Na, folic acid) as an independent variable, and the subgroup (each in a separate model) as a covariate. ^b^ The interaction “Product and Subgroup” was studied and reported. ^c^ Low levels of each biomarker were defined as levels ≤ median; high levels were defined as levels > median. The medians were; plasma total folate = 14.2 nmol/L, RBC-folate = 633 nmol/L, vitamin B12 = 254 pmol/L, tHcy = 7.5 µmol/L, and Hb = 15.1 g/L (for men) and 13.4 g/L (for women).

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
