# Peer review of "Pharmacokinetics of Sodium and Calcium Salts of (6S)-5-Methyltetrahydrofolic Acid Compared to Folic Acid and Indirect Comparison of the Two Salts"

_nutrients, 2020, doi:10.3390/nu12123623_

Round 1
Reviewer 1 Report
This is an interesting manuscript that compares the kinetics of plasma 5-methylTHF after equivalent doses of folic acid and 5-methylTHF. Their results have biological relevance in that the 5-methyl derivative appears to produce a better “folate” profile than that produced by folic acid (more 5-methyl and less unchanged folic acid in the plasma). They also show no basic difference between the Ca salt (previous study) and the Na salt (their study) in terms of folate kinetics.
MAJOR COMMENTS
1. The authors do a variety of statistical tests and data manipulations to try to show the equivalence between their studies with the Na salt and those previously done with the Ca salt. I am convinced by these arguments. However, they do not really discuss any biological differences in the subjects between the two studies (other than the obvious sex differences). The main concern here is the difference in the AUC of 17 nmol/L*h between the two folic acid groups in the two studies (which is a big difference percentage-wise). I realize that they examined a correction factor (ie subtraction of 17 from all values) and showed that there was no differences when using that factor. While this is statistically comforting, it would be good to hear from the authors why they think there was a difference between the two reference groups in these studies, and more importantly, whether there was a biological difference in the two reference groups that could explain the math.
2. In the text (lines 254-255), the authors state that the total folate in plasma in Figure 2 after 5 methyl was “stronger” than that after folic acid. The actual data in Figure 2B shows only a slightly larger total folate between the two groups (none of which are probably statistically different). Suggest revising the text.
3. Text lines 271-280 and Table 3. The authors remove the Tmax data for 2 subjects that were apparent outliers, with the result that the Tmax became shorter after 5methyl than after folic acid. This seems OK, but I would be curious why they didn’t remove the same two outliers from the Cmax data and re-analyze it. The actual Cmax values of those two may not have been “outlying” compared to the other Cmax values, but since they were so unusual in terms of Tmax, this does question their Cmax values also.
MINOR COMMENTS
1. Supplement table 2 – there is a superscript c in the AUC row that is not explained in the footnotes.
2. Line 223 – I think they mean that the AUCs of the reference plasma folic acid in 2009 was higher than that in 2019, which is why they then subtract 17 from all the AUCs as a correction factor. The reason I suggest this wording change is that 17 is the difference in AUCs for folic acid in the 2009 and 2019 studies.
3. In line 144, it is stated that there were 23 women in the 2009 study, but in line 201, it is stated as 24 women. Please clarify. Related to this in line 239, it is stated that 3 women were excluded from the 2009 study because of “implausible” data. What is meant by that – and how do they (the other authors?) validate the exclusion?
4. Line 249 – error! Reference source not found??
5. Line 284 – Table 4 not 5.
6. Line 415. Instead of moreover, I would use however or in contrast, because in the subsequent sentence they are indicating a lesser effect of 5methyl compared to folic acid, while prior to the sentence, 5methyl was stated as being advantageous.
Reviewer 2 Report
This is a small cross-over clinical trial comparing the kinetics of folate absoprtion between folic acid and a sodium salt of methyl folate. The design mimics a previous kinetic study comparing the absorption of a calcium salt of methyl folate to folic acid.
The findings are incremental and not particularly novel, with the new information relating priomarily to the use of a sodium salt of methylfolate. The authors find that a single dose has a statistically significant and more rapid rise in plasma 5-methylfolate than folic acid. This is not surprising and of arguable clinical importance.
The claims that there are serious safety concerns over folic acid are overblown. 400 mg FA is safe and efficacious and inexpensive and has been used for decades. There are concerns over excessive intake, particularly in the face of b12 deficiency; however, that is only marginally relevant to the very limited conditions tested in this study (a single dose of FA and a single dose of 5MTHF). No safety conclusions can be drawn here and comments to that effect should be eliminated.
The focus on UPLC/MS results obscures the important finding that the final Total Folate concentrations and AUC are of a similar magnitude even if they are statistically different in total when presented as AUC (Fig 1 and Table 2). The total rise of unmetabolized folate in plasma is very small compared to the difference presented between the plasma rise of methyl folate for the methylfolate salt vs folic acid and the large differences only persist for about 2 hours. (Figure 1). [The figure fails to indicate statistical differences between the groups and doesn't indicate what the error bars represent]. In other words, if one adds 5MTHF + FA it does not equal Total Folate. However, Total Folate is the clinically interpretable parameter and the clinical difference between the treatments is hard to discern. The long term impact of this difference would have to be demonstrated in a separate clinical trial with a different design and claims of superiority in this regard are speculative. Indeed, the gap between FA and 5MTHF in plasma kinetics might indicate that FA is more readily assimilated by tissues as it is reduced to tetrahydrofolate (THF) that is polyglutamated and trapped in the cells, whereas 5MTHF is a monoglutamate that enters and leaves the cell freely until it is converted into THF in the tissues and polyglutamated. The almost statistically significant interaction with low B12, showing that individuals with B!2 below the median had a higher rise in plasma 5MTHF supports this interpretation. However, there are many other hypothetical explanations. In short, the observations are noteworthy but they should be presented objectively and not overinterpreted.
The major flaw in the paper is the combination of a report on a primary research study with a secondary review and re-analysis of prior independent studies on calcium-MTHF. This analysis has serious limitations` this is not the place to discuss them and the entire comparison does not belong in this paper. If the authors wish to publish the paper they should focus solely on the present trial and write a separate analysis comparing the two studies.
Reviewer 3 Report
Obeid et al. describe interesting features regarding the abilities of (6S)-5-Methyltetrahydrofolic acid [(6S)-5-Methyl-THF] salts and folic acid to rise (6S)-5-Methyl-THF levels in plasma.
The authors limited the analysis to a short range of time (0-8hrs). Based on the results obtained, it would be interesting investigate the effect of 6S)-5-Methyltetrahydrofolic acid [(6S)-5-Methyl-THF] salts in longer treatment and in relation of disease such as MTHFR polymorphisms.
The manuscript is very well written, but the following minor changes are required.
Introduction
I believe would be very Interesting for the readers of Nutrients, to know in which foods the folates are present, and after the absorption of (6S)-5-Methyltetrahydrofolic acid [(6S)-5-Methyl-THF] salts (Ca and Na) what are the beneficial effects for the human body. I suggest to the authors to include this aspect in the introduction.
Lane (45) “Folates are a source of purine and pyrimidine nucleotides, one-carbon units, and methyl groups”
This sentence can be a very good introduction, it is informative for the biological role of folates in cells but need to be adjusted. Folates are not a source of nucleotide, but a source of purine and pyrimidine bases required in nucleotide synthesis then implicated in DNA, RNA structures .. Folates donate one- carbon group during the synthesis of purines, formylmethionyl-tRNA, thymidylate, serine, and methionine. Folates are also micronutrients and essential donors of methyl groups.. Please, can the author explain better this part?
Lane (27 and 244) “Twenty-four adults (50% women)”
“50% woman” does not have much sense .. I would change with “12 men and 12 woman.”
Supplementary Figure 2
What does it mean “Washout approx. two weeks”?
I believe that washout is not the right term to use in this case, but why “approximately”? Are 14 days or not? Please change or clarify.
Figure 2
I would add the legends in the graph that are intuitive.
Supplemental Figure 8
Why the authors decided to show this figure in supplementary material? A consistent part of the present study is focused on (6S)-5-Methyltetrahydrofolic acid [(6S)-5-Methyl-THF] salts, and I think the authors should change the Supp Figure 8 in main figures 4.
Lane (355) “Supplementation of folate in young women aims..”
Please, can the authors define “young women”. I would simply change it with the age range..
Lane (from 390 to 393)
I think the authors highlight a very interesting point, but I am worried there are not much evidence. This sentence cannot be accepted as it is.. “High doses of folic acid could even aggravate the extremely low rate of conversion of folic acid by exceeding the DHFR-binding capacity and thereby blocking and inhibiting the enzyme’s active sites” can the author add at least one reference in the end of this sentence regarding the effect of high doses of folic acid inhibiting DHFR activity? If they cannot, please remove “could” from the sentence, and explain in the same sentence that it is just a hypothesis.
“Inhibition of DHFR causes disruption of nucleotide biosynthesis and DNA replication which in turn leads to cell death.” In this case the authors citate a book section (reference 29) which does not show any experiment that confirm the impaired DNA replication by the effect of DHFR inhibition. I strongly recommend the authors to add at least another reference that show how Inhibition of DHFR (even through inhibitors, small molecules or similar) leads to disruption of DNA replication, otherwise change the sentence.
Lane (from 434 to 436)
The sentence is incomplete, these findings are confirmed only for a short time treatment. Please include this info in the sentence.
Reviewer 4 Report
This is a well planned and executed study answering an important question in the field regarding folate supplementation. As the authors have shown, unmetabolized folic acid is a small component of the dose and this is cleared or further converted following absorption or excreted.While replacing folic acid with methylfolate may be appropriate, its stability is an issue and would need addressing. Methylfolate may be appropriate for supplementation during pregnancy and for treatment of specific disorders needing folate. With daily folic acid supplementation through food fortification is likely to show some small amount of unmetabolized folic acid. It is highly unlikely that it has any pathologic effects or metabolic significance.
Author Response
no changes are required by reviewer 4
Round 2
Reviewer 2 Report
My concerns in the previous review have not been adequately addressed.
Claiming "superiority" of methylfolate over folic acid because it raises plasma methylfolate to a statistically significant amount more than folic acid does, is neither proof of biological efficacy or physiological benefit. A rise in circulating folate could just as well indicate harm or certain physiological adverse conditions, such as the methylfolate trap, rather than benefit. The concerns or open questions relating to potential harms of excess folic acid do not relate only to masking of pernicious anemia but possibly to adverse outcomes relating to high total folate in general.
This paper makes a small modest contribution to the literature with straightforward observations on a single dose of sodium salt of methylfolate in a small number of healthy adults.
There is no place to argue for clinical benefits. There is also no place to conflate a primary research paper, with an additional secondary analysis of previously published data. That belongs in a second paper where they can review the literature and express their speculative opinion on the potential merits and limitations of the quasi-new compound that they are developing.
In short, the framing, discussion and conclusions overstate the case and go beyond what can be supported by a straightforward and objective discussion of the present study's experimental results.
